# Whole Exome Sequencing and In Silico Analysis of Human Sertoli in Patients with Non-Obstructive Azoospermia

**DOI:** 10.3390/ijms232012570

**Published:** 2022-10-20

**Authors:** Hossein Azizi, Danial Hashemi Karoii, Thomas Skutella

**Affiliations:** 1Faculty of Biotechnology, Amol University of Special Modern Technologies, Amol 46156-64616, Iran; 2Institute for Anatomy and Cell Biology, Medical Faculty, University of Heidelberg, Im Neuenheimer Feld 307, 69120 Heidelberg, Germany

**Keywords:** non-obstructive azoospermia, Sertoli cells, microarray, infertility, genes expression

## Abstract

Non-obstructive azoospermia (NOA) is a serious cause of male infertility. The Sertoli cell responds to androgens and takes on roles supporting spermatogenesis, which may cause infertility. This work aims to enhance the genetic diagnosis of NOA via the discovery of new and hub genes implicated in human NOA and to better assess the odds of successful sperm extraction according to the individual’s genotype. Whole exome sequencing (WES) was done on three NOA patients to find key genes involved in NOA. We evaluated genome-wide transcripts (about 50,000 transcripts) by microarray between the Sertoli of non-obstructive azoospermia and normal cells. The microarray analysis of three human cases with different non-obstructive azoospermia revealed that 32 genes were upregulated, and the expressions of 113 genes were downregulated versus the normal case. For this purpose, Enrich Shiny GO, STRING, and Cytoscape online evaluations were applied to predict the functional and molecular interactions of proteins and then recognize the master pathways. The functional enrichment analysis demonstrated that the biological process (BP) terms “inositol lipid-mediated signaling”, “positive regulation of transcription by RNA polymerase II”, and “positive regulation of DNA-templated transcription” significantly changed in upregulated differentially expressed genes (DEGs). The BP investigation of downregulated DEGs highlighted “mitotic cytokinesis”, “regulation of protein-containing complex assembly”, “cytoskeleton-dependent cytokinesis”, and the “peptide metabolic process”. Overrepresented molecular function (MF) terms in upregulated DEGs included “ubiquitin-specific protease binding”, “protease binding”, “phosphatidylinositol trisphosphate phosphatase activity”, and “clathrin light chain binding”. Interestingly, the MF analysis of the downregulated DEGs revealed overexpression in “ATPase inhibitor activity”, “glutathione transferase activity”, and “ATPase regulator activity”. Our findings suggest that these genes and their interacting hub proteins could help determine the pathophysiologies of germ cell abnormalities and infertility.

## 1. Introduction

Sertoli cells (SCs) are important in the physiology and pathophysiology of mammalian testes [1]. SCs are the first somatic cells to differentiate in the testes in the embryo and are hypothesized to guide future testis development. SCs undergo the differentiation process during puberty (about 14 days old in mice), which involves stopping proliferation, protein expression and transcription changes, and functional maturation [2]. Mature SCs form the blood–testis barrier (BTB) to establish microenvironments for spermatogenesis and release various functional products to feed germ cells and coordinate spermatogenesis activities [3,4]. SCs, in particular, secrete a variety of molecules (including *glial cell line-derived neurotrophic factor (GDNF)*, *stem cell factor (SCF), fibroblast growth factor 2 (FGF2), and bone morphogenic protein 4 (BMP4))*, which establish and maintain the balance between SC self-renewal and differentiation [5]. Consequently, any changes in the populations or functions of SCs result in defective spermatogenesis and, ultimately, sterility. SCs are important targets for regulating spermatogenesis [6]. In mammals, spermatogenesis is governed by a complex regulatory process that includes hormones, growth factors (transforming growth factor-beta (TGF-β), tumor necrosis factor-alpha (TNF), and Glial cell line-derived neurotrophic factor (GDNF), endotoxins, and proinflammatory cytokines. Recent research indicates that the Sertoli cell is required for spermatogenesis and that mutations in these genes may disrupt spermatogenesis and induce infertility [7,8].

Infertility affects approximately 15% of couples, with males being the main cause of the couple’s infertility approximately one-third of the time. Causes of infertility in men include hormonal disorders, physical problems, lifestyle problems, etc. Azoospermia is classified as obstructive Azoospermia (OA) or non-obstructive Azoospermia (NOA), each having very different etiologies [9,10]. NOA is male infertility caused by spermatogenic dysfunctions of testicular tissue. Patients with NOA can produce a small amount of sperm, but not enough for fertility. In NOA patients, the structure of the seminiferous tubules of the testis is altered, the maturation of spermatogenic cells is impaired, and spermatogenic cell meiosis is stopped [11,12]. A recent study has shown that the spermatogenesis problem involves both testicular histopathology and hormone abnormalities [13].

Whole exome sequencing (WES) has identified several genes in NOA pedigrees, including *DNA meiotic recombinase 1, stromal antigen 3, testis expressed 11, shortage in chiasmata 1, synaptonemal complex central element protein 1, meiosis-specific with an OB-fold coiled-coil domain containing 155, testis expressed 14, testis expressed 15,* and *X-ray repair cross-complementing 2* [14,15]. This study found Sertoli cell gene expression deficiencies that are involved in spermatogenesis.

However, it is unclear whether WES is necessary during fertility in vitro, and some studies showed on-stage association in the human seminiferous epithelium [16]. Lastly, there are a few studies on this whole-genome sequencing in male germ cells. In this experimental study, we analyzed the expressions of the whole genome sequencing genes in Sertoli cells by microarray and in silico analyses.

## 2. Results

### 2.1. Isolation of Sertoli Cells

Microarrays from various cell groups were initially evaluated on the gene expression profiles for analysis compared to the normal cases. Figure 1 summarizes the different groups, lab experiments, and in silico analyses.

### 2.2. Up- and Downregulated Gene Expressions between Non-Obstructive Azoospermia and a Normal Case by Microarray

We evaluated whole sequencing (about 50,000 transcripts) by microarray. The microarray analysis of three human cases with different non-obstructive azoospermia revealed that 32 genes were upregulated, and the expressions of 113 genes were downregulated versus the normal case (Figure 2 and Figure 3). The microarray analysis of three human cases with different non-obstructive azoospermia revealed that stromal membrane-associated protein (SMAP-1), lines homolog 1 (LINS1), SERTAD4 antisense RNA 1 (Also known as C1orf133), zinc finger protein 865 (ZNF865), chromobox 7 (CBX7), kirre-like nephrin family adhesion molecule 3 (KIRREL3), SELM, ATPase phospholipid transporting 9A (ATP9A), phosphatase and tensin homolog (PTEN), ribose 5-phosphate isomerase A (RPIA), casein kinase 1 epsilon (CSNK1E), Josephin domain containing 1 (JOSD1), zinc finger protein 518B (ZNF518B), CAP-Gly domain containing linker protein family member 4 (CLIP4), F-box protein 3 (FBXO3), transmembrane protein 119 (TMEM119), lysophosphatidic acid acyltransferase 1 (ICT1), single stranded DNA binding protein 2 (SSBP2), phosphatidylinositol 3-kinase catalytic subunit type 3 (PIK3C3), sphingolipid transporter 3 (SPNS3), small nucleolar RNA, C/D box 56B (SNORD56B), outer mitochondrial membrane lipid metabolism regulator 3 (OPA3), tubulin tyrosine ligase-like 9 (TTLL9), cystatin 9 (CST9), WIBG, ring finger protein 216 pseudogene 1 (RNF216L), clathrin heavy chain (CLTC), purine rich element binding protein B (PURB), lysophospholipase 2 (LYPLA2), Translin (TSN), arbonic anhydrase 11 (CA11) and LysM domain containing 4 (LYSMD4) were upregulated, and expressions of PMS2L1, CHEK2, TRIP13, and POLD4 were upregulated versus the normal case (Figure 3).

The microarray analysis of three human cases with different non-obstructive azoospermia showed that microtubule-associated protein 4 (MAP4), ubiquitin conjugating enzyme E2 E3 (UBE2E3), aminopeptidase puromycin sensitive (NPEPPS), endothelin receptor type A (EDNRA), suppression of tumorigenicity 7 (ST7), ubiquitin-like with PHD and ring finger domains 1 (UHRF1), transmembrane channel-like 6 (TMC6), ribosomal modification protein rimK-like family member B (RIMKLB), X-linked inhibitor of apoptosis (XIAP), patatin-like phospholipase domain containing 5 (PNPLA5), ribonucleotide reductase regulatory subunit M2 (RRM2), ring finger protein 114 (RNF114), ring finger and WD repeat domain 3 (RFWD3), zinc finger E-box binding homeobox 1 (ZEB1), sorting nexin 4 (SNX4), SPG7 matrix AAA peptidase subunit (SPG7), methylenetetrahydrofolate dehydrogenase (NADP+ dependent) 1-like (MTHFD1L), histone deacetylase 6 (HDAC6), coiled-coil domain containing 144A (CCDC144A), transmembrane O-mannosyltransferase targeting cadherins 4 (TMTC4), exportin 6 (XPO6), kelch-like family member 12 (KLHL12), E2F associated phosphoprotein (EAPP), zinc finger protein 254 (ZNF254), FKBP prolyl isomerase 5 (FKBP), ribosome biogenesis homolog (URB2), poly(ADP-ribose) polymerase 2 (PARP2), small nucleolar RNA, H/ACA box 16B (SNORA16B), serine incorporator 4 (SERINC4), protein kinase C and casein kinase substrate in neurons 2 (PACSIN2), anillin, actin binding protein (ANLN), neurolysin (NLN), replication factor C subunit (RFC3), FTO alpha-ketoglutarate dependent dioxygenase (FTO), YEATS domain containing 4 (YEATS4), signal transducer and activator of transcription 1 (STAT1), L-2-hydroxyglutarate dehydrogenase (L2HGDH), leucyl-tRNA synthetase (LARS), phosphatidylinositol glycan anchor biosynthesis class K (PIGK), WD repeat domain 90 (WDR90), G2 and S-phase expressed 1 (GTSE1), PC4 and SFRS1 interacting protein 1 (PSIP1), telomerase associated protein 1 (TEP1), ADP ribosylation factor-like GTPase 17B (ARL17), GTP binding protein 10 (GTPBP10), polypeptide N-acetylgalactosaminyl transferase 15 (GALNTL2), U2AF homology motif kinase 1 (UHMK1), 3-oxoacid CoA-transferase 1 (OXCT1), glutathione S-transferase theta 2 (GSTT2), tuftelin interacting protein 11 (TFIP11), hydroxysteroid 17-beta dehydrogenase 7 (HSD17B7), stathmin 1 (STMN1), ribonuclease P/MRP subunit p30 (RPP30), pyruvate dehydrogenase phosphatase regulatory subunit (PDPR), non-SMC condensin I complex subunit G (NCAPG), MYC associated zinc finger protein (MAZ), HEAT repeat containing 2 (HEATR2), activator of basal transcription 1 (ABT1), ALMS1 centrosome and basal body associated protein ALMS1), sperm antigen with calponin homology and coiled-coil domains 1-like (CYTSA), sestrin 1 (SESN1), RAB2B, latent transforming growth factor beta binding protein 3 (LTBP3), folliculin interacting protein 2 (FNIP2), RecQ-like helicase 5 (RECQL5), establishment of sister chromatid cohesion N-acetyltransferase 2 (ESCO2), RAP2A, major histocompatibility complex, class I-related (MR1), synaptojanin 2 (SYNJ2), folliculin interacting protein 1 (FNIP1), protein phosphatase 2 regulatory subunit B’delta (PPP2R5D), regulator of G protein signaling 2 (RGS2), rhomboid domain containing 2 (RHBDD2), polypyrimidine tract binding protein 3 (PTBP3: Also known as ROD1), signal regulatory protein gamma (SIRPG), cyclin Y (CCNY), MLLT11 transcription factor 7 cofactor (MLLT11), ATA-box binding protein associated factor 9b (TAF9B), RAS p21 protein activator 2 (RASA2), kinesin family member 4A (KIF4A), MAS related GPR family member F (MRGPRF), glutathione S-transferase theta 2B (GSTT2B), EBP-like (EBPL), signal regulatory protein beta 1 (SIRPB1), serpin family B member 8 (SERPINB8), PWWP domain containing 2A (PWWP2A), dihydrofolate reductase (DHFR), codanin 1 (CDAN1), phospholipid scramblase 3 (PLSCR3), zinc finger protein 253 (ZNF253), spectrin beta, non-erythrocytic 1 (SPTBN1), WD repeat domain 19 (WDR19), Rac family small GTPase 2 (RAC2), zinc finger protein 618 (ZNF618), sorbitol dehydrogenase (SORD), Family with sequence similarity 118 member A (FAM118A), kinetochore-associated Ndc80 complex subunit SPC24 (SPC24), INO80 complex subunit C (INO80C), PMS1 homolog 1 (PMS1), carboxylesterase 1 (CES1), Ras related GTP binding D (RRAGD), phosphatase domain containing paladin 1 (Also known as KIAA1274), progestin and adipoQ receptor family member 4 (PAQR4), MID1, and S100 calcium binding protein A1 (S100A1) were upregulated versus the normal case.

### 2.3. Group Comparison and Protein Class Sorting

A comparison of all the non-obstructive azoospermia and normal cell groups together revealed that 145 transcripts (32 genes upregulated and 113 downregulated transcripts) were differentially expressed (Appendix A). The analysis of transcripts (using PANTHER (Protein ANalysis THrough Evolutionary Relationships, Foster City, CA, USA) showed that differentially expressed RNAs covered diverse gene sequences localized all around the cell, including organelles, membranes, extracellular matrix, and cell junctions (Figure 1). The PANTHER server showed that upregulated genes were involved in metabolite interconversion enzyme (22.20%), protein modifying enzyme (16.70%), gene-specific transcriptional regulator (16.70%), transporter (11.1%), membrane traffic protein (5.6%), chaperone (5.6%), cell adhesion molecule (5.6%), DNA metabolism protein (5.6%), cytoskeletal protein (5.6%) and translational protein (5.6%). Moreover, the PANTHER server showed that downregulated genes were involved in metabolite interconversion enzyme (21.2%), protein modifying enzyme (12.9%), protein-binding activity modulator (11.8%), gene-specific transcriptional regulator (9.4%), DNA metabolism protein (8.2%), cytoskeletal protein (7.1%), RNA metabolism protein (5.9%), transmembrane signal receptor (4.7%), scaffold/adaptor protein (3.5%), transporter (3.5%), translational protein (2.4%), extracellular matrix protein (2.4%), defense/immunity protein (2.4%), chromatin/chromatin-binding, or regulatory protein (1.2%), calcium-binding protein (1.2%), transfer/carrier protein (1.2%) and chaperone (1.2%) (Figure 4).

### 2.4. Construction of Differentially Expressed Gene–Protein–Protein Interaction(PPI) Network

The PPI network was constructed through the STRING database (Version: 11.5), and the software Cytoscape (Version: 3.8.2) was used to visually analyze the PPI network graph. The CytoHubba plug-in MCC algorithm (cytoHubba: Version 0.1) was used to screen out the up/downregulated genes, and they were intersected with 50 differential genes in the genes regulatory network, and finally, 11hub target genes—KLF4A, CLTC, STMN1, ANLN, MKI67, GTSE1, NCAPG, UHRF1, SPC24, RRM2, and RFC3—were obtained. Through the analysis of differentially expressed genes, it was found that KLF4A is the core target gene and exists in the gene regulatory network (Figure 5).

### 2.5. Biological Process, Molecular Functions, and Cellular Components of Enrichment Analysis

The Enrich tool analysis revealed that upregulated Differentially Expressed Genes (DEGs) were enriched in three GO terms, while downregulated DEGs were incorporated with three GO terms. Functional enrichment analysis demonstrated that the biological process (BP) terms “inositol lipid-mediated signaling” (GO:0048017) (*p* < 0.001), “positive regulation of transcription by RNA polymerase II” (GO:0045944) (*p* < 0.0001), and “positive regulation of DNA-templated transcription” (GO:0045893) (*p* < 0.0001) were significantly overexpressed in upregulated DEGs (Figure 6).

Functional enrichment analysis demonstrated that the BP investigations of downregulated DEGs highlighted “mitotic cytokinesis” (GO:0000281) (*p* < 0.001), “regulation of protein-containing complex assembly” (GO:0043254) (*p* < 0.002), “cytoskeleton-dependent cytokinesis“ (GO:0061640) (*p* < 0.004), and “peptide metabolic process“ (GO:0006518) (*p* < 0.006) (Figure 6) as significantly overexpressed in downregulated DEGs.

Overrepresented molecular function (MF) terms included “ubiquitin-specific protease binding” (GO:1990381) (*p* < 0.0001), “protease binding” (GO:0002020) (*p* < 0.001), “phosphatidylinositol trisphosphate phosphatase activity” (GO:0034594) (*p* < 0.001), and “clathrin light chain binding” (GO:0032051) (*p* < 0.001) (Figure 6). Interestingly, the MF analysis of downregulated DEGs revealed that “ATPase inhibitor activity” (GO:0042030) (*p* < 0.001), “glutathione transferase activity” (GO:0004364) (*p* < 0.001), and “ATPase regulator activity” (GO:0001671) (*p* < 0.001) (Figure 6) were overexpressed in upregulated DEGs.

Cellular component (CC) terms included “phosphatidylinositol 3-kinase complex” (GO:0005942) (*p* < 0.0001), “autolysosome” (GO:0044754) (*p* < 0.001), “trans-Golgi network membrane” (GO:0032588) (*p* < 0.001), “mitotic spindle” (GO:0072686) (*p* < 0.001), and “clathrin coat” (GO:0030118) (*p* < 0.001). CC terms in downregulated DEGs included “intracellular non-membrane-bounded organelle” (GO:0043232) (*p* < 0.001), “mitotic spindle microtubule” (GO:1990498) (*p* < 0.001), “microtubule” (GO:0005874) (*p* < 0.001), and “endosome membrane” (GO:0010008) (*p* < 0.01) (Figure 6).

### 2.6. Isolated and Selected Candidate MicroRNAs

In this section, after identifying 15 genes, LARS, RRM2, SPTBN1, UHMK1, STAT1, ABT1, DHFR, ZNF260, SORL1, NPEPPS, OXCT1, RPP30, MID1, KLHL12, and FNIP1, we isolated and selected the most relevant microRNAs (Figure 7). Accordingly, hsa-miR-4764-3p, hsa-miR-3201, hsa-miR-4791, hsa-miR-4787-3p, hsa-miR-1278, hsa-miR-3713, hsa-miR-337-5p, hsa-miR-4633-5p, and mmu-miR-344 were observed more clearly than other microRNAs (Figure 7). These microRNAs are candidates for up- and downregulation of LARS, RRM2, SPTBN1, UHMK1, STAT1, ABT1, DHFR, ZNF260, SORL1, NPEPPS, OXCT1, RPP30, MID1, KLHL12, and FNIP1 genes (Figure 7).

## 3. Discussion

Sertoli cells are testicular somatic cells that are required for testis development and spermatogenesis [1]. Sertoli cells support the advancement of germ cells in spermatozoa by direct interaction and by controlling the environmental microenvironment inside the seminiferous tubules. Sertoli cells have been shown to consume glucose aerobically at a high rate and release lactate and pyruvate; lactate and pyruvate are needed for germ cell maintenance [17,18]. In non-obstructive azoospermia, we show that the genes pyruvate dehydrogenase phosphatase regulatory (PDPR) and sorbitol dehydrogenase regulatory (SORD) were decreased (downregulated).

Furthermore, Sertoli cells contained branched-chain amino acid aminotransferase activity, amino acid, lipid, and carbohydrate metabolic processes, and 15% to 30% of the 4-methyl-2-oxopentanoate formed by transamination of leucine was released, taken up by germ cells, and converted to 2-hydroxy-4-methylvalerate [18]. Furthermore, Grootegoed et al. demonstrated that, in addition to glucose and fatty acids, the direct oxidation of glutamine and leucine might provide a significant portion of the energy needed by Sertoli cells [19,20]. In non-obstructive azoospermia, ribose 5-phosphate isomerase A (RPIA) gene expression increased (upregulation), while ribosomal modification protein rimK-like family member B (RIMKLB), L-2-hydroxyglutarate dehydrogenase (L2HGDH), and hydroxysteroid 17-beta dehydrogenase 7 (HSD17B7) gene expression decreased (downregulation). We hypothesize that abnormalities in these genes may lead to infertility.

Proteolytic enzymes, which are produced and released by Sertoli cells in the testicular seminiferous tubule, play a key role in spermatogenesis [21,22]. Only little proteolytic activity was detected in Sertoli cell-conditioned supernatants, suggesting that the proteases under investigation were mostly found on Sertoli cell membranes [23]. The peptide hormones utilized in this research were discovered to be involved in the endocrine, paracrine, or autocrine control of testicular cells [24]. As a result, the membrane-associated proteases described here might be implicated in the metabolism and deactivation of these peptides [25,26]. In non-obstructive azoospermia, the gene ATPase phospholipid transporter 9A (ATP9A) was upregulated, whereas the ring finger and WD repeat domain 3 (RFWD3) and phosphatidylinositol glycan anchor biosynthesis class K (PIGK) were downregulated. We hypothesize that abnormalities in these genes prevent Sertoli cells and spermatogenesis from synthesizing and secreting these proteins.

The ubiquitin–proteasome system (UPS) is critical for the precise regulation of protein homeostasis, maintaining the efficiency of certain protein groups at particular stages and their inactivation beyond that stage [27,28]. Many UPS components have been shown to influence spermatogenesis progression at various stages [27]. Novel testis-specific proteasome isoforms have been found as required and distinct for spermatogenesis, particularly in recent years [29]. Diverse USP components play roles in mammalian spermatogenesis, involving (1) the composition of proteasome isoforms at each stage of spermatogenesis [6]; (2) the specificity of each proteasome isoform and the associated degradation events [27]; (3) the E3 ubiquitin ligases mediating protein ubiquitination in male germ cells [30]; and (4) the deubiquitinases involved in spermatogenesis and male fertility [31]. Our microarray analysis revealed that Josephin domain containing 1 (JOSD1), F-box protein 3 (FBXO3), and ring finger protein 216 pseudogene 1 (RNF216L) were upregulated, while ubiquitin-conjugating enzyme E2 E3 (UBE2E3), ubiquitin-like with PHD and ring finger domains 1 (UHRF1), ring finger protein 114 (RNF114), ring finger, and WD repeat domain were downregulated. We hypothesize that altering the expressions of these ubiquitin–proteasome system genes may alter proteasome isoforms and affect protein homeostasis.

The blood–testis barrier (BTB) in mammals creates a unique microenvironment for germ cell growth and maturation in order to sustain and maintain spermatogenesis [32]. The BTB is located between Sertoli cells and germ cells, as well as between germ cells and Sertoli cells. Sertoli cells are metabolically active, dynamic epithelial cells that support the seminiferous epithelium [33]. Microtubules in these cells are organized in linear arrays parallel to the cell’s long axis, producing a cage-like structure surrounding the nucleus. The Sertoli cell form varies depending on the needs of the growing germ cells. A-kinase anchor proteins (AKAPs) are microtubule dynamics regulators that contribute to appropriate BTB functioning [34]. Akap9 deletion has been linked to major microtubule architecture alterations in Sertoli cells and BTB loss in Akap9 null animals [35].

Although Sertoli cell polarity is critical for effective spermatogenesis, it is unclear when and how these cells become polarized [36]. Rho GTPase signaling has been linked to cell polarity and other cellular processes (e.g., cell cycle, cell migration, cytoskeletal structure) in a variety of biological situations [36,37]. CDC42, a key member of the Rho GTPase family, has been linked to various aspects of cellular biology, including the establishment of *polarity in epithelial* cells [38]. CDC42 interacts with and activates multiple effector proteins when activated in GTP-bound form. CDC42 has been linked to the control of Sertoli–Sertoli BTB dynamics throughout the seminiferous epithelial cycle and Sertoli-germ cell contacts at the apical ES mediated by the effector protein IQGAP1 [36,39]. However, little is known about CDC42’s involvement in Sertoli cells during testicular differentiation and spermatogenesis in vivo.

Furthermore, male germ cell meiosis and morphological changes related to spermatozoa formation add intricacy to this process that is not seen in other self-renewing tissues [40,41]. Biological links between developing germ cells may be crucial for their survival, and these interactions explain a species’ constant germ cell cycle dynamics [42,43]. We discovered that some cell cycle and cytokinesis genes, such as RRM2 (ribonucleotide reductase regulatory subunit M2), HDAC6 (histone deacetylase), ring finger and WD repeat domain 3 (RFWD3), E2F associated phosphoprotein (EAPP), G2 and S-phase expressed 1 (GTSE1), U2AF homology motif kinase 1 (UHMK1) and stathmin 1 (STMN1) were downregulated in non-obstructive azoospermia. We hypothesize that these genes might influence the Sertoli cell cycle. Sertoli cell cycle defects may prevent sperm from maturing.

It was recently suggested that genomic instability might result in poor overall health and infertility. Infertility or poor sperm quality might be among the disadvantages or indications of genomic instability [44]. Genome instability may be to blame for severe instances of infertility in men—a multifactorial condition exhibited, among other things, by altered spermatogenesis [45]. Male infertility research has produced data to support the concept that poor spermatogenesis (especially in idiopathic infertility patients) is often related to variables other than testicular dysfunction [46,47,48]. Our findings show that WD repeat domain 90 (WDR90), telomerase associated protein 1 (TEP1), non-SMC condensin I complex subunit G (NCAPG), the establishment of sister chromatid cohesion N-acetyltransferase 2 (ESCO2), kinesin family member 4A (KIF4A), PWWP domain containing 2A (PWWP2A), and INO80 complex subunit C were downregulated in Sertoli of non-obstructive azoospermia. We hypothesize that these genes could malfunction during the replication, transcription, and translation processes.

The cytoskeleton is a structural component of all eukaryotic cells. It provides structural support and functional pliability, playing critical roles in many essential processes like motility, intracellular trafficking, differentiation, and cell division [49,50]. As a result, several important differentiations and spermatogenic processes are regulated by cytoskeletal dynamics [51,52]. Our findings reveal that the genes TTLL9, WDR90, ANLN, SPTBN1, KIF4A, PACSIN2, and STMN1 are up/downregulated in Sertoli cells. We emphasize the vital relevance of a dynamic and properly controlled cytoskeleton for male infertility. Furthermore, we hypothesize that Sertoli cells with abnormalities in these genes are unable to differentiate and divide throughout the spermatogenesis and sperm maturation processes. The development of technologies, such as CRISPR, is also expected to contribute to providing meaningful data for proving several postulated processes.

## 4. Material and Methods

### 4.1. Patient and Control Selection

Three patients with Azoospermia caused by defective spermatogenesis and three control cases were evaluated (Table 1). They were scheduled for testicular biopsy for genetic investigations and afterward for intracytoplasmic injection of the oocyte. The testicular biopsies and sperm extraction were not timed to coincide with the oocyte retrieval and intracytoplasmic sperm injection (ICSI) cycles, which took place 4–5 months later. Cryopreserved testicular spermatozoa were recovered. The University of Heidelberg Ethics Committee authorized the project. Serum samples were collected and analyzed in accordance with World Health Organization guidelines. After no spermatozoa were identified in the pellet produced from semen centrifuged at 1500 g for 12 min, Azoospermia was diagnosed. The developmental, social, medical, and reproductive histories, as well as the history of urological procedures and gonadotoxic exposure, were all reported. Each patient was subjected to a general, systemic, and genital examination. A Prader orchidometer was used to calculate testicular volume. The enzyme-linked immunoassay was used to measure the plasma concentrations of the follicle-stimulating hormone (FSH) (normal value: 2–3 IU/L) and luteinizing hormone (LH) (normal value: 3–10 IU/L). Radioimmunoassay was used to measure plasma testosterone (normal range: 9.4–37.0 nmol/L). The coefficients of variation within and between assays did not surpass 6.5%. Prior to testicular biopsy, each patient had a chromosomal analysis of a venous blood sample performed using conventional procedures. None of the patients had the Y-chromosome microdeletion test done. A cascade screening of common CF gene mutations was performed to aid in the exclusion of patients with obstructive Azoospermia when genital cystic fibrosis (CF) was suspected due to poor sperm volume, acidic pH, or the lack of vasa deferentia. Azoospermic individuals with retrograde ejaculation, genital tract blockage, or endocrine problems were excluded from the research. Only individuals with Azoospermia related to faulty spermatogenesis were investigated.

### 4.2. Surgical and Sample Processing

The surgical procedure was performed by removing a substantial portion of testicular tissue by a transverse incision of the albuginea, either equatorially or in the cranial section of the testis. The pieces were washed in a human tubal fluid medium to remove the blood before being handed over to the biologist for microscopic analysis. Only after that were the testicular tissue surfaces irrigated with the Ringer solution (containing 75 mg gentamicin/100 mL for antisepsis). After that, hemostasis was achieved by gently pushing the testicular tissue for 3 min with gauze moistened with the aforesaid antiseptic solution, followed by extremely restricted and cautious bipolar microcoagulation. To avoid discomfort and tunica vaginalis adhesions, the tunica vaginalis opening was fixed with a continuous vicryl 5/0 after 1.5 mg betamethasone was instilled into the vaginalis cavity. Each piece of testicular tissue was inserted in a sterile Petri plate with 0.75 mL of a sperm washing medium and distributed by isolating individual seminiferous tubules using glass slides and mincing individual tubules. We centrifuged the diluted semen sample at 300–500 g for 5–10 min. Each tissue sample was counted to determine the existence and quantity of spermatozoa. When the fluid recovered following centrifugation of the suspension contained more than 100 spermatozoa/mm^3^, the viability of the extracted spermatozoa was assessed using the eosin–nigrosin live/dead stain test. Figure 1 and Table 1 show the experimental design for this investigation. The microarray study initially examined the gene expression patterns of the microarray from several groups of cells.

### 4.3. Selection of Human Sertoli Cells

From the cryopreserved testicular tissue, the dissociated tubules were enzymatically digested for 30 min at 37 °C with 750 U/mL collagenase Type IV (Sigma, St Louis, MO, USA), 0.25 g/mL dispase II (Roche, Ludwigsburg, Germany), and 5 g/mL DNase in HBSS buffer with Ca++ and Mg++ (PAA) to generate a single-cell suspension. The digestion was then stopped with 10% ES cell-qualified FBS. The cell suspension was centrifuged for 15 min at 1000 rpm after passing through a 100 m cell strainer. After removing the supernatant, the pellet was rinsed with HBSS solution containing Ca++ and Mg++. After washing, the cells (approximately 2 × 105 cells per cm^2^) were plated into five culture dishes (*d* = 10 cm) coated with 0.2% gelatin (Sigma), into hGSC (human germ stem cell) medium consisting of StemPro hESC medium, 1% N2-supplement (Invitrogen, Waltham, MA, USA), 6 mg/mL D+ glucose (Sigma), 5 g/mL bovine serum albumin (Sigma), 1% L-glutamine (Sigma). Cells in this culture media were cultured for 96 h in a CO_2_-incubator at 37 °C and 5% CO_2_ in the air. After 72 h, the culture medium volume of 5 milliliters was changed with new culture media of the same volume, and the cells were grown for another 4 days. The culture media was carefully removed on day 7, and the testis cell culture was gently washed with 5 mL DMEM/F12 culture medium with L-glutamine (PAA) per plate to harvest the germ cells adhered to the monolayer of adherent somatic cells connected to the dish bottom. Pipetting 5 mL of DMEM/F12 culture media was used to repeat this process [53,54,55,56].

### 4.4. Sertoli Cell Isolation

According to our previous research, human Sertoli cells were isolated and identified [17]. In brief, testicular tissues were washed three times before being cut into 0.2 cm pieces and treated with Enzyme I (10 mL of DMEM containing 2 mg/mL type IV collagenase and 10 mg/mL DNase I) for 15 min at 34 °C. After that, seminiferous tubules were washed again and incubated for 10–15 min at 34 °C with En-zyme II (4 mg/mL collagenase IV, 2.5 mg/mL hyaluronidase, 2 mg/mL trypsin, and 10 mg/mL DNase I). Tissue blocks were filtered through a 40-mm cell strainer, and germ cell and Sertoli cells were grown in DMEM/F-12 supplemented with 10% FBS at 34 °C in 5% CO_2_ at 34 °C [53,57,58]. The Sertoli cells were selected from the cultures from pooled cell groups by a cell culture surface panning strategy with laminin followed by lectin binding. Non-laminin binding cells were transferred to a lectin-coated dish. The attached cells were selected for Microarray studies. Our experiments selected highly pure Vimentin+, Sox9+, Gata4+,Vasa-, and UTF1-human Sertoli cell cultures [59].

### 4.5. Collection of Single Cells from the Population of the Enriched Human Sertoli Cell Population with a Micromanipulation System

The lectin-sorted Sertoli cells in each sample were washed with a culture medium and re-suspended before being transferred to a single cell suspension on top of a tiny culture plate. The dish’s top was put on a pre-warmed (37 °C) working platform of a Zeiss inverted microscope equipped with a micromanipulation system. A micromanipulation pipette was used to harvest the cells step by step at a magnification of 20×. The characteristic morphology of short-term cultivated spermatogonia was clearly visible. This was largely due to their round form, diameter of 6–12 m, and high nucleus-to-cytoplasm ratio, which was seen as a distinct, glittering cytoplasmic ring between the round nucleus and the outer cell membrane.

#### RNA Extraction and WES Preparation

Total RNA isolated for WES was prepared using the RNeasy Mini Kit (Qiagen, Germantown, MD, USA) followed by an amplification step with MessageAmp aRNA Kit (Ambion). Samples (for analysis) were provided to the microarray facility at the University Clinic, Tübingen, Germany. Gene expression analysis was performed using the Human U133 + 2.0 Genome oligonucleotide array (Affymetrix, Santa Clara, CA, USA). The raw data were provided to MicroDiscovery GmbH, Berlin, Germany, for normalization and biostatistical analysis.

### 4.6. Microarray Analysis

Total RNA was extracted from lectin-selected Sertoli using the RNeasy Mini Kit (Qiagen), and then amplified using the MessageAmp aRNA Kit (Ambion, Austin, TX, USA). Moreover, 100 hundred cells were harvested per probe in each sample using the micromanipulation equipment, transferred immediately into 10 L of RNA direct lysis solution, and kept at –80 °C. The RNA was isolated from cell samples and microarray datasets obtained from the hybridization of human mRNA-derived cDNA after amplification using Super-Amp^TM^ technology. Agilent Whole Human Genome Oligo Microarrays 8 × 60 K v^2^ (Miltenyi, Germany) were analyzed by bioinformatics tools.

### 4.7. Microarray Data Normalization and Analyses

R Statistical Environment version 3.6.2 was used to analyze microarray data (16 December 2021). The Bioconductor software affy version 1.28.0 was used to condense the data. The condensation criteria were as follows: bg.correct = FALSE, normalize = FALSE, and pmcorrect = FALSE. summary.method = ‘medianpolish’, method = ‘pmonly’ A multi-dimensional variant of the Lowess normalization approach was used to achieve further normalizing between samples. Parts of the data were entered into the IPA Ingenuity tool to assess gene functions and pathways.

### 4.8. Data Processing and Differential Expression Analysis

We used a microarray to evaluate the expression of 50,000 transcripts in non-obstructive azoospermia. These genes were then compared to normal cells. The default settings were adj. value 0.05 and |log2 fold change (FC)| > 2. The volcano map and heat map were used to visually describe DEGs. The volcanic map and heat map were created using R software.

### 4.9. Group Comparison and Protein Class Sorting

Gene selection and gene ontology: the online program ArrayMining was utilized to compare the research groups for DEGs. The resulting gene list was then loaded into PANTHER (http://www.pantherdb.org/; accessed on 16 June 2021), an online tool for gene ontology analysis.

### 4.10. Gene ontology (GO) Investigation

Enrich (http://amp.pharm.mssm.edu/Enrichr/; the Ma’ayan Lab, sinai, Egypt; accessed on 16 June 2021), an online software tool for functional gene annotation, was used to study the KEGG (Kyoto encyclopedia of genes and genomes, https://www.genome.jp/kegg/; San Francisco, CA, USA; accessed on 17 June 2021) and Reactome (https://reactome.org/; University Ave, Toronto, ON, Canada; accessed on 17 June 2021) enrichment pathway. To confirm the biological roles of the genes involved in the PPI network of the first protein–protein interaction nodes with the RNA sequence, we performed a functional gene enrichment analysis using the STRING (https://string-db.org/; San Francisco, CA, USA, accessed on 4 June 2021) enrichment analysis in Cytoscape software. The biological process mediated by related infertility genes was highlighted by the ShinyGO tool.

### 4.11. Target Transcript Protein–Protein Interaction Network Construction

The STRING database is utilized for protein–protein interaction (PPI), and the Cytoscape (https://cytoscape.org/; Boston, MA, USA; accessed on 16 June 2021) program is used for visual analysis of the PPI network to identify the association between candidate target genes further. CytoHubba uses the degree value to indicate the size of the node. The plug-in MCC algorithm selects the top 50 up/downregulated core genes.

### 4.12. Target Transcript Prediction and miRNA Regulatory Network Construction

The TargetScan (https://www.targetscan.org/vert_80/; Whitehead Institute for Biomedical Research, Cambridge, UK; accessed on 20 June 2021), miRTarBase (https://mirtarbase.cuhk.edu.cn/ Institute of Bioinformatics and Systems Biology, Neuherberg, Germany; accessed on 20 June 2021), and StarBase (https://starbase.sysu.edu.cn/; Hertford, UK; accessed on 20 June 2021) databases were used to identify the target genes of differentially expressed genes and miRNAs. We searched for the intersection of target genes predicted by all three databases and microarray differential genes to find the candidate target miRNA. The regulatory link between miRNA and mRNA was used to build the miRNA–mRNA regulatory network. The miRNA–mRNA regulatory network was visualized using Cytoscape software (version 3.9.2).

## 5. Conclusions

Finally, the up/downregulation of these genes can cause NOA infertility. These genes are involved in the metabolite interconversion enzyme, protein modifying enzyme, protein-binding activity modulator, gene-specific transcriptional regulator, DNA metabolism protein, cytoskeletal protein, RNA metabolism protein, transmembrane signal receptor, scaffold/adaptor protein, transporter, translational protein, extracellular matrix protein, defense/immunity protein, chromatin/chromatin-binding, regulatory protein, calcium-binding protein, transfer/carrier protein, and chaperone. Our findings suggest that these genes and their interacting hub proteins could help determine the pathophysiology of germ cell abnormalities and infertility.

## Figures and Tables

**Figure 1 ijms-23-12570-f001:**
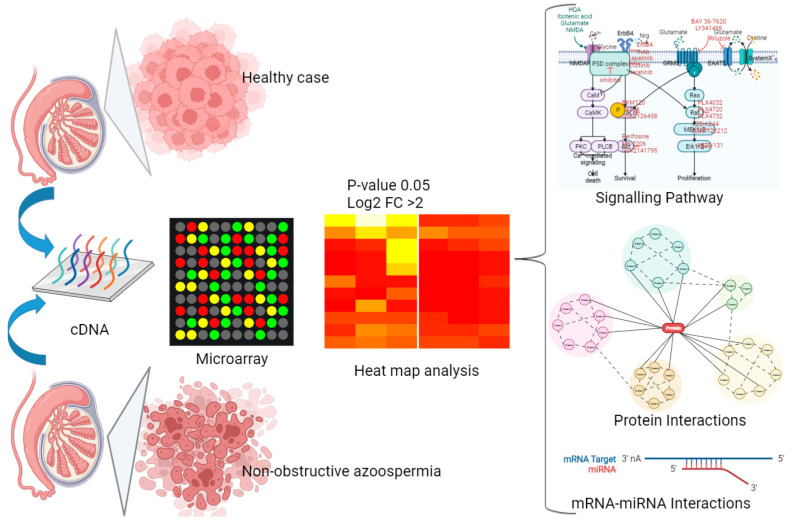
Diagram of the experimental design (groups, lab experiment, and in silico analysis).

**Figure 2 ijms-23-12570-f002:**
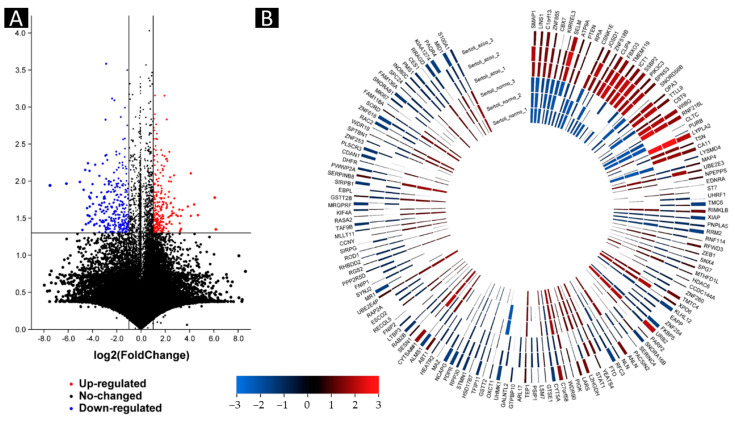
Differentially expressed gene testis samples in patients with non-obstructive azoospermia and healthy cases. (**A**) Volcano map of the DEGs. Red dots represent upregulation, blue dots represent downregulation, and gray dots represent no differential expression. (**B**) Heat map of DEGs. Red represents upregulation, and blue represents downregulation.

**Figure 3 ijms-23-12570-f003:**
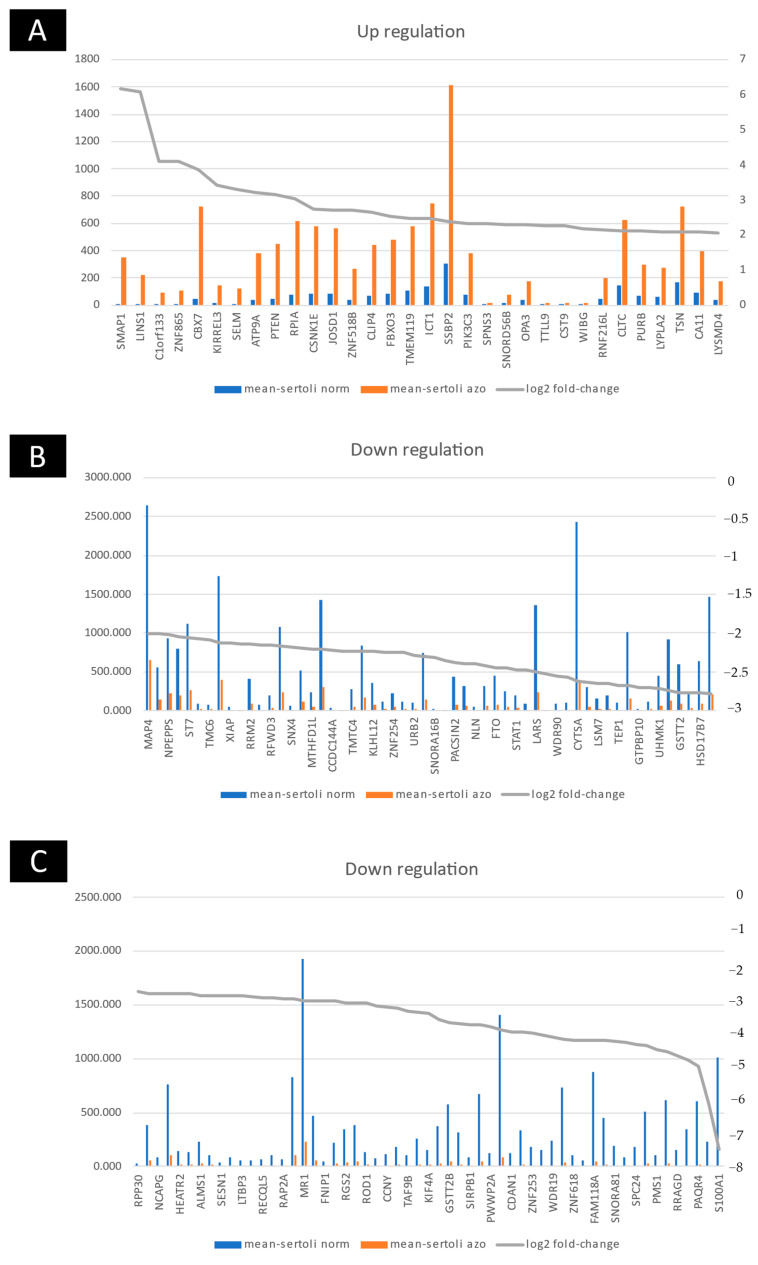
The ratio of up/downregulated genes with non-obstructive azoospermia and normal cells. (**A**) Upregulated gene and (**B**,**C**) downregulated gene.

**Figure 4 ijms-23-12570-f004:**
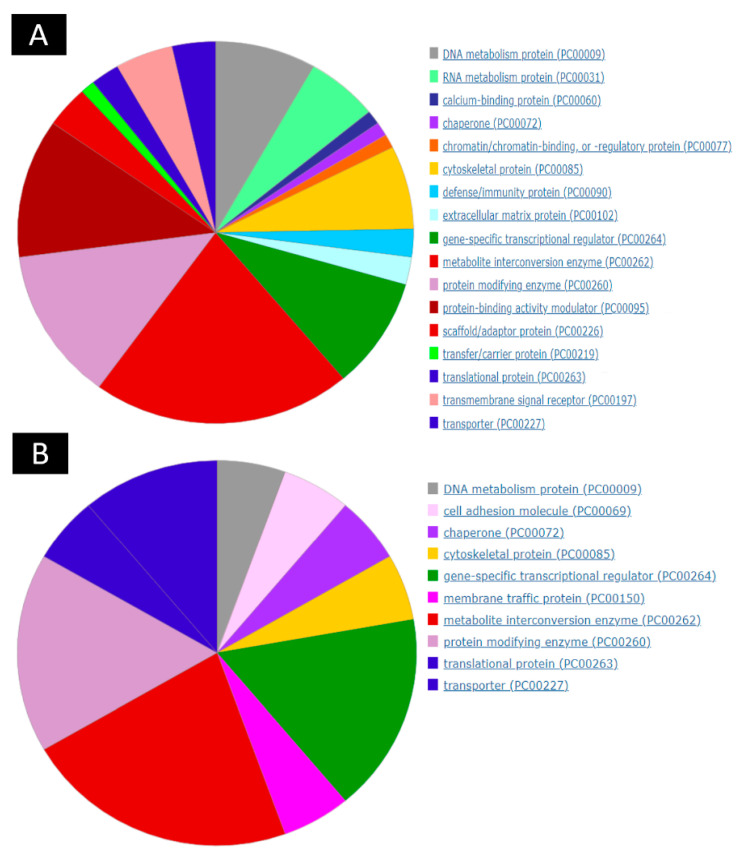
Group comparison and protein class analysis. The PANTHER analysis of the most significant DEGs (to classify them into protein classes). (**A**) PANTHER analysis of up-regulation genes (**B**) PANTHER analysis of down-regulation genes.

**Figure 5 ijms-23-12570-f005:**
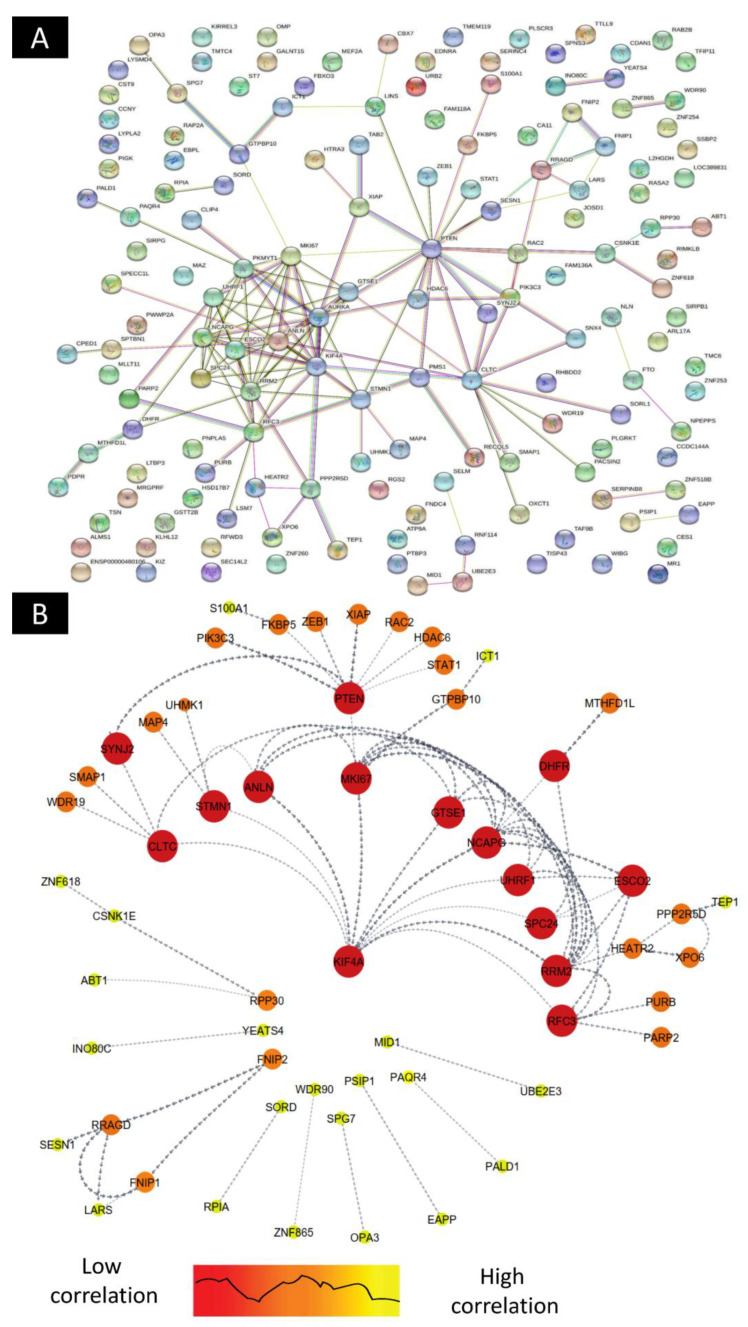
STRING protein–protein interaction network based on gene interaction. (**A**) PPI network up/downregulation and related genes, and (**B**) direct linkage of genes correlated in up/downregulated genes.

**Figure 6 ijms-23-12570-f006:**
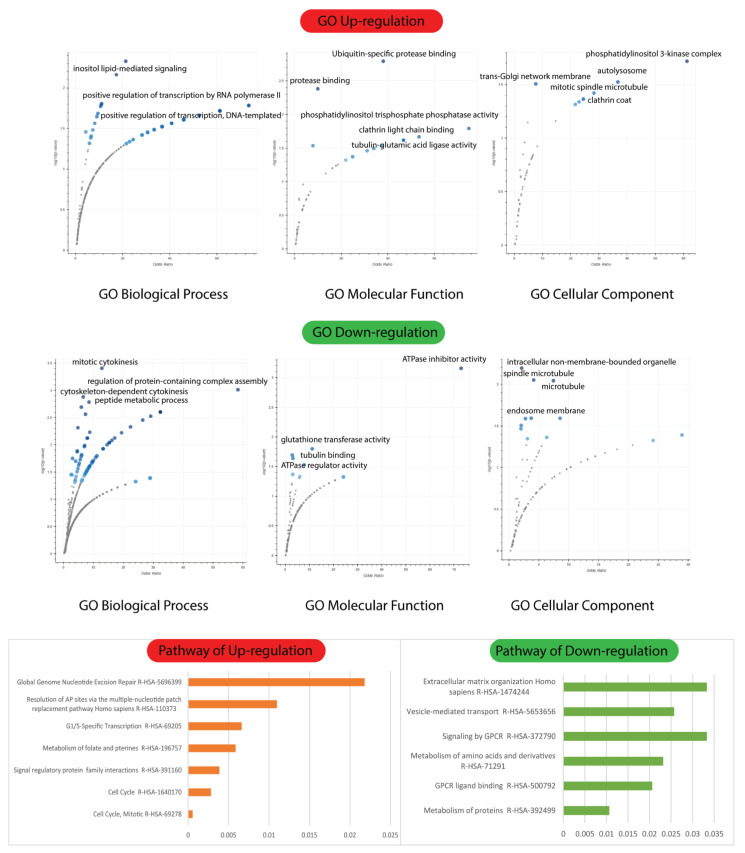
GO biological process, molecular function cellular component, and signaling pathways of up/downregulated genes with non-obstructive azoospermia and normal cells.

**Figure 7 ijms-23-12570-f007:**
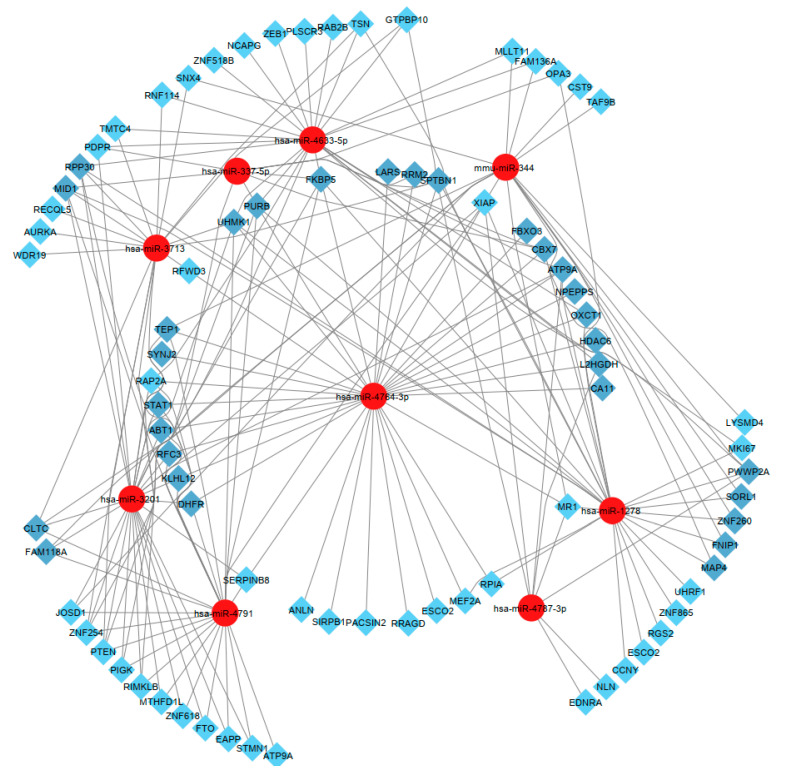
Hub genes and hub target miRNA. The up/downregulated genes intersected with nine differential miRNAs in the miRNA-gene regulatory network.

**Table 1 ijms-23-12570-t001:** Characteristics of patients and sperm samples included in this study.

Patient (Sample)	Indication	Histopathological Diagnosis	Sperm Quality
Patient 1	Non-obstructive Azoospermia	Hypospermatogenesis	Rare, Concentration: 60 *×* 106 spermatozoa/mL Motility: 70% motile Morphology: 31% normal
Patient 2	Non-obstructive Azoospermia	Hypospermatogenesis	Rare, Concentration: 200 *×* 106 spermatozoa/mL Motility: 50% motile Morphology: 48% normal
Patient 3	Non-obstructive Azoospermia	Hypospermatogenesis	Rare, Concentration: 80 *×* 106 spermatozoa/mL Motility: 50% motile Morphology: 26% normal
Normal	Normozoospermia	/	Abundant, motile spermatozoa
Normal	Normozoospermia	/	Abundant, motile spermatozoa
Normal	Normozoospermia	/	Abundant, motile spermatozoa

## Data Availability

The original contributions presented in the research are included in the article/Appendix A; further inquiries can be directed to the corresponding author.

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
