# Peer review of "Whole Exome Sequencing and In Silico Analysis of Human Sertoli in Patients with Non-Obstructive Azoospermia"

_ijms, 2022, doi:10.3390/ijms232012570_

Round 1

Reviewer 1 Report

The authors looked into the role of Sertoli cells in Non-obstructive azoospermia (NOA). 

The manuscript is hard to follow and understand. 

Some inconsistencies  or need better explanation:

Manuscript talks about Whole exome sequencing was done on 3 NOA patients. Materials methods did not explain anything about WES.  

The manuscript talks about 50000 genes. It is debatable, I am assuming authors worked on 50000 transcripts? 

In section 2.2: authors mentioned 32 gene upregulated and 113 downregulated genes. section 2.3  talks about 50000 transcripts differentially expressed? 

The manuscript needs a better explanation or hypothesis. When two conditions were taken most likely there would be a differential expression. Authors are unable to connect the dots between different biological processes and NOA.

Author Response

Dear reviewer,

Thanks for taking the time to carefully review our manuscript entitled “Whole exome sequencing and in-silico analysis in human Sertoli of patients with non-obstructive Azoospermia”, the comments are illuminating for our present and further work.

Now we have revised according to your comments, and the list of changes or a rebuttal against each individual point which is being raised as follows:

Sincerely,

Hossein Azizi (Corresponding author), [email protected]

Thomas Skutella (Corresponding author), [email protected]

The authors looked into the role of Sertoli cells in Non-obstructive azoospermia (NOA). 

The manuscript is hard to follow and understand. 

Some inconsistencies or need better explanation:

Manuscript talks about Whole exome sequencing was done on 3 NOA patients. Materials methods did not explain anything about WES.  

Reply: We add them as “RNA extraction and WES prepare:” section in material and method.

The manuscript talks about 50000 genes. It is debatable, I am assuming authors worked on 50000 transcripts? 

Reply: Thank you so much. We evaluated genome-wide transcripts (about 50000 transcripts) by microarray between Sertoli of non-obstructive azoospermia and normal cells

In section 2.2: authors mentioned 32 gene upregulated and 113 downregulated genes. section 2.3  talks about 50000 transcripts differentially expressed? 

Reply: We revised them. One hundred forty-five transcripts were differentially expressed.

The manuscript needs a better explanation or hypothesis. When two conditions were taken most likely there would be a differential expression. Authors are unable to connect the dots between different biological processes and NOA.

Reply: We attempted to revise them. After finding DEG, we used many databases to confirm and improve explanations like signaling pathways, gene ontology, protein interactions,miRNA candidates etc.

Reviewer 2 Report

Ijms-1914265

“Whole exome sequencing and in-silico analysis in human Sertoli of patients with non-obstructive Azoospermia”

In the present work, the authors performed whole exome sequencing on three NOA patients in order to find key genes involved in NOA. Although the number of NOA patients is very small, the work is very innovative and represents a solid approach to the condition. I suggest the improvement of the following minor points.

Page 2, Line 2: SSC means SCs? Please revise

Page 2. Please review the structure of the following phrase: “Patients with NOA either cannot produce only a small amount of sperm”. The structure is confused and needs to be revised.

Page 2. Please revise the phrase “A recent study has shown that the spermatogenesis problem involves both localized and diverse organs”, in order to make it clearer. What do authors intend to say?

Page 2. WES: please write unabbreviated before using abbreviation

Page 7. What is the reference of the present work at the Ethical Committee?

Page 8. At Table 1, some doubts arise. how could sperm of patient 1 possess “abundant and motile spermatozoa”? Revise accordingly. No reference is given to testicular sperm morphology.

At Table 1 the first row (black) is not filled.

If ejaculates are classified according to WHO Manual it is surprising that at the last row; related to the normal patient number 3, normal morphology is quite low for a normozoospermic ejaculate. Please clarify.

On the clinical evaluation of the patients, the testicular volume is measured and some hormonal levels as FSH, LH and testosterone. Are these results showed? Did they evidence any abnormality? Please clarify.

Page 8. The description of the surgical technique is complete. In fact, the collection of samples at testicular level could itself induce inflammation and can aggravate the NOA condition. How could this be avoided?

Page 8. Please add some information on the centrifugation conditions (time and speed)

Page 8. Please revise the sentences, as at its present form are confused:

“Suppl. shows the experimental design for this investigation. Table 2.

Author Response

Dear reviewer,

Thanks for taking the time to carefully review our manuscript entitled “Whole exome sequencing and in-silico analysis in human Sertoli of patients with non-obstructive Azoospermia”, the comments are illuminating for our present and further work.

Now we have revised according to your comments, and the list of changes or a rebuttal against each individual point which is being raised as follows:

Sincerely,

Hossein Azizi (Corresponding author), [email protected]

Thomas Skutella (Corresponding author), [email protected]

In the present work, the authors performed whole exome sequencing on three NOA patients in order to find key genes involved in NOA. Although the number of NOA patients is very small, the work is very innovative and represents a solid approach to the condition. I suggest the improvement of the following minor points.

Page 2, Line 2: SSC means SCs? Please revise

Reply: We revise it.

Page 2. Please review the structure of the following phrase: “Patients with NOA either cannot produce only a small amount of sperm”. The structure is confused and needs to be revised.

Reply: We revise the sentences.

Page 2. Please revise the phrase “A recent study has shown that the spermatogenesis problem involves both localized and diverse organs”, in order to make it clearer. What do authors intend to say?

Reply: We revise the sentences.

Page 2. WES: please write unabbreviated before using abbreviation

Reply: Done.

Page 7. What is the reference of the present work at the Ethical Committee?

Reply: The University of Heidelberg and the Amol University of Special Modern Technologies (Ir.ausmt.rec.1400.05) Ethics Committee were approved.

Page 8. At Table 1, some doubts arise. how could sperm of patient 1 possess “abundant and motile spermatozoa”? Revise accordingly. No reference is given to testicular sperm morphology.

Reply: Thanks. We revise them.

At Table 1 the first row (black) is not filled.

Reply: Done.

If ejaculates are classified according to WHO Manual it is surprising that at the last row; related to the normal patient number 3, normal morphology is quite low for a normozoospermic ejaculate. Please clarify.

Reply: Thank you so much. We misplaced the last row between normal and Patient. We revised them.

On the clinical evaluation of the patients, the testicular volume is measured and some hormonal levels as FSH, LH and testosterone. Are these results showed? Did they evidence any abnormality? Please clarify.

Reply: For better separation of normal and patient, we just used hormonal levels (FSH, LH etc.) in normal cases to confirm. We did not use hormonal levels (FSH, LH etc.) in patient cases.

Page 8. The description of the surgical technique is complete. In fact, the collection of samples at testicular level could itself induce inflammation and can aggravate the NOA condition. How could this be avoided?

Reply: Testicular samples were directly frozen after biopsy. Therefore induction of inflammation directly related to the fast surgical procedure is very unlikely.

Page 8. Please add some information on the centrifugation conditions (time and speed)

Reply: Sperm wash for diluted semen sample is centrifuged at 300-500g for 5-10 minutes. We add it to the manuscript.

Page 8. Please revise the sentences, as at its present form are confused:

“Suppl. shows the experimental design for this investigation. Table 2.

Reply: We corrected it.
